# Prioritizing Management of Non-Native Eurasian Watermilfoil Using Species Occurrence and Abundance Predictions

**Alison Mikulyuk [1,*], Catherine L. Hein [1], Scott Van Egeren [2], Ellen Ruth Kujawa [3] and M. Jake Vander Zanden [4]**

1.  Bureau of Water Quality, Division of Environmental Management, Wisconsin Department of Natural Resources, 101 S Webster Street., Madison, WI 53707, USA; Catherine.Hein@wisconsin.gov
2.  Bureau of Water Quality, Division of Environmental Management, Wisconsin Department of Natural Resources, 107 Sutliff Avenue, Rhinelander, WI 54501, USA; Scott.Vanegeren@wisconsin.gov
3.  Bureau of Science Services, Wisconsin Department of Natural Resources, 2801 Progress Road, Madison, WI 53716, USA; ellen.kujawa@gmail.com
4.  Center for Limnology, University of Wisconsin-Madison, 680 N Park Street, Madison, WI 53706, USA; mjvanderzand@wisc.edu
*   Correspondence: Alison.Mikulyuk@Wisconsin.gov

**Abstract:** Prioritizing the prevention and control of non-native invasive species requires understanding where introductions are likely to occur and cause harm. We developed predictive models for Eurasian watermilfoil (EWM) (*Myriophyllum spicatum* L.) occurrence and abundance to produce a smart prioritization tool for EWM management. We used generalized linear models (GLMs) to predict species occurrence and extended beta regression models to predict abundance from data collected on 657 Wisconsin lakes. Species occurrence was positively related to the nearby density of vehicle roads, maximum air temperature, lake surface area, and maximum lake depth. Species occurrence was negatively related to near-surface lithological calcium oxide content, annual air temperature range, and average distance to all known source populations. EWM abundance was positively associated with conductivity, maximum air temperature, mean distance to source, and soil erodibility, and negatively related to % surface rock calcium oxide content and annual temperature range. We extended the models to generate occurrence and predictions for all lakes in Wisconsin greater than 1 ha (N = 9825), then prioritized prevention and management, placing highest priority on lakes likely to experience EWM introductions and support abundant populations. This modelling effort revealed that, although EWM has been present for several decades, many lakes are still vulnerable to introduction.

**Keywords:** species distribution model; SDM; species abundance; Eurasian watermilfoil; EWM; Myriophyllum spicatum; non-native species; invasive species; aquatic plants; aquatic macrophytes

---

## 1. Introduction

Non-native species are a leading driver of global environmental change. They can alter ecosystem structure and function and decrease global biodiversity [1–4]. They are economically costly and can pose hazards to human health [5,6]. Invasive species have been recorded from over half the extant phyla and divisions, and their modes of impact are as diverse as the invaders themselves [7–10]. Thus, the vulnerability of any given area to invasive species is a central concern for ecologists and natural resource managers.

Assessing lake-specific invasion vulnerability requires understanding three filters that mediate species invasions. The first filter relates to likelihood of species introduction, the second includes the probability that the introduced species establishes a self-sustaining population, and the third relates to its likely impact [11]. In other words, lakes in which a non-native species is likely to arrive, survive, and thrive are considered vulnerable. Predicting which lakes are vulnerable is helpful for management; it allows the informed and efficient application of limited resources for prevention and control, thereby minimizing adverse ecological and economic effects.

Predicting whether a lake is vulnerable first requires predicting where that species is likely to occur. Species occurrence is mediated by the first two invasion filters related to a species' ability to arrive and survive. Important factors to consider include the potential for propagules to disperse, climate, and the availability of critical resources; these are reviewed elsewhere [12]. By mathematically relating species occurrences and spatial and environmental characteristics within a known sample of lakes, we may predict species distributions for the population [13]. Including dispersal-related factors is especially important when a species range is changing, as is the case for many non-native species, but dispersal is not routinely accounted for in species distribution models [14–16].

The second critical step in assessing lake-specific invasion vulnerability is to determine whether the species is likely to have adverse effects. The effects of non-native species are generally more difficult to forecast than their distributions and, as a result, they are not often predictively modelled [17–19]. A promising solution is to use abundance as a proxy for impact because it is more tractably modelled as a function of explanatory variables [14]. For most invasive species, abundance and impact are positively related, although the precise shape of that relationship can take different forms [19–22].

The central goal of this study was to describe lake-specific vulnerability to Eurasian watermilfoil (EWM), a non-native invasive macrophyte. We extended prior work to predict EWM occurrence by adding an empirical abundance model [23–25]. We predicted species occurrence and abundance as a function of water quality, land use, dispersal potential, geology, and climate variables. Predictors were selected for their link to EWM environmental suitability or aquatic invasive species (AIS) occurrence [24,26–28]. We then united predictions of occurrence and abundance in a prioritization framework to identify lakes at risk for developing abundant EWM populations. We separated vulnerable lakes into tiers of increasing management priority and thereby offer a simple tool to prioritize prevention efforts and management actions that reduce the impact of EWM.

## 2. Materials and Methods

### 2.1. Invasion History of Eurasian Watermilfoil

Eurasian watermilfoil is an invasive aquatic plant that can grow to high abundance in certain freshwater systems. Native to Europe, Asia, and North Africa, the precise date of introduction to the United States is unknown but probably occurred between 1880 and 1940 [29]. It has since spread throughout the continental US and Canada [30]. In Wisconsin, it is present in over 800 waterbodies and has steadily expanded its range northward following introduction in the south-central region during the 1960s [31]. When at nuisance levels, EWM forms thick mats at the surface that prevent navigation, reduce property values, and affect native species [32,33].

Eurasian watermilfoil is a costly and time-consuming management challenge [34]. From 2018 to 2020, over half of Wisconsin's total funding for Aquatic Invasive Species control grants supported EWM control, amounting to over $500,000 USD annually. This figure excludes additional private expenditures for prevention, planning, and control. Eurasian watermilfoil management can also have significant non-target effects. Common management tools such as herbicides and mechanical harvesting have been associated with negative ecological effects on native plant communities [35–37]. Given the high economic and ecological cost of the species and its management, preventing the spread of EWM to uninvaded lakes is a priority. However, containing the species to the more than 1000 locations in which it is found in Wisconsin, or shielding the other 15,000 lakes from future introduction are both efforts

that vastly exceed available funding. We used predictive modelling to identify vulnerable lakes where EWM is likely to be introduced and become abundant. We hope our findings will increase the efficiency of prevention and control funding, focusing it on the most vulnerable uninvaded waterbodies.

*2.2. Occurrence and Abundance of Eurasian Watermilfoil*

We used aquatic plant survey data obtained from the Wisconsin Department of Natural Resources (WDNR) to generate EWM occurrence and abundance models. WDNR staff and partners working under a long-term aquatic plant management research and monitoring program conducted standardized and repeatable plant surveys on a total of 657 waterbodies distributed across Wisconsin's three lake-rich ecoregions. Lake surface area in the survey dataset ranged from 1.36 to 3958 ha, while watershed land use ranged from pristine to nearly entirely developed [38]. Surveyors performed their work during growing seasons from 1 May to 1 October 2005–2016, employing a grid-based point-intercept survey method to observe species' presence/absence at many sites per lake. The sampling window was selected to minimize the variation in abundance over time during the growing season and thus reduce the influence of seasonality on abundance estimates (unpublished data). The number of points sampled per lake ranged from 32 to 4149, with a mean of 406 points [39]. We estimated species-specific abundance as littoral frequency of occurrence in the littoral zone using the following method. At each sampling point, observers used a double-sided bow rake attached to a 4.5 m pole to collect macrophytes from a ~0.3 m$^2$ area. A similar rake head attached to a rope was used to collect macrophytes from sites deeper than 4.5 m [40]. All live macrophytes detached by the rake were identified to species [41,42]. We then defined littoral zones per lake based on sampled depths that were equal to or shallower than the 99th percentile of ordered depths at which macrophytes were observed. On average, 234 sample points fell within lake littoral zones. For each species, we calculated abundance as the number of occurrences divided by the total number of littoral points in each lake. To reduce the number of false absences resulting from sparse EWM populations that were below the plant survey's limit of detection, we augmented the dataset's presence/absence observations using a list of verified population records obtained from the WDNR. Verified records are confirmed by management staff who review observations originating from several sources, including citizen reports, routine monitoring by WDNR staff, and formal Aquatic Invasive Species detection surveys. This procedure added presence observations for 92 surveyed lakes with a 0 for EWM abundance, resulting in a total number of 385 lakes with verified populations out of a total of 657 surveyed.

*2.3. Explanatory Variables*

Our goal was to predict EWM occurrence and abundance on lakes statewide. We compiled information on factors thought to predict EWM occurrence and abundance if they were available for at least 7000 of the 9285 Wisconsin lakes greater than 1 ha surface area. Predictors represented a suite of factors related to water quality and lake morphometry as well as regional variables related to dispersal, land use, geology, and climate (Table S1, Supplementary Materials). We used watersheds delineated by Menuz et al. to calculate variables expressed at the watershed scale [43]. We extracted climate variables at lake centroids from WorldClim (worldclim.org). We described watershed geological characteristics using the whole-rock percentage of calcium oxide (CaO) in near-surface geology and soil erodibility (K-factor) [44,45]. High surface rock calcium oxide content relates positively to surface water conductivity, acid neutralizing capacity, and calcium content, whereas highly erodible soils are linked to increased runoff, sedimentation, and nutrient loading [44,46]. We calculated percent agriculture (crops and pasture) and percent urban land use in the watershed [47]. As a proxy for dispersal potential or propagule pressure, we computed the mean distance (km) between a lake's centroid and all other Wisconsin lakes with EWM [48]. Factors related to vector pressure included nearby road density and lake surface area [26,49]. We calculated the density of vehicle roads (m/m$^2$) in a 500 m buffer around each lake [50] and lake surface area extracted from the 24k hydrography dataset and maximum depth from the WDNR Register of Waterbodies [51,52]. Spatial analyses were

conducted using the R packages 'rgdal' (v. 1.2–5), 'raster' (v. 2.5–8), 'sp' (v. 1.2–4), and 'rgeos' (v. 0.3–22) and ArcGIS (v. 10.2.2, Environmental Systems Research Institute, Redlands, CA, USA) [53–56]. Finally, a comprehensive database of limnological parameters provided water conductivity (µS/cm), alkalinity (mg $CaCO_3$), pH, and satellite-estimated Secchi depth (m) for Wisconsin lakes [57].

Missing-at-random values comprised less than 3% of all observations. We imputed missing variables using predictive mean matching and the package 'mice' (v. 2.30) over 50 iterations [58]. We log-transformed highly skewed numeric variables and square-root-transformed skewed percentages (see Table 1). We computed variance inflation factors (VIFs) for each variable in the dataset using the function 'vif' in the package 'car' (v. 2.1–4) [59]. We sequentially excluded variables with the largest inflation factor until no inflation factor exceeded 10.

**Table 1.** Estimated coefficients for Eurasian watermilfoil (EWM) occurrence model developed using data from 657 plant survey lakes. Coefficients expressed as odds ratios calculated for centered and scaled predictors, profile confidence intervals in parentheses. Negative relationships are indicated by odds ratios less than 1.

| Predictor | Coefficient |
|---|---|
| Intercept | 0.13 *** (0.07−0.21) |
| Road density (log (m/ha +1)) | 1.93 *** (1.42−2.74) |
| Surface area (log ha) | 1.72 *** (1.33−2.31) |
| Maximum air temp. (°C × 10) | 1.69 ** (1.23−2.40) |
| Maximum depth (log m +1) | 1.55 ** (1.20−2.06) |
| Conductivity (log µS/cm) | 1.47 (0.72−3.20) |
| Alkalinity (log mg $CaCO_3$ +1) | 1.44 (0.67−3.03) |
| Soil erodibility (kwfact) | 1.17 (0.93−1.49) |
| Watershed urban ( $\sqrt{\%}$ ) | 1.07 (0.77−1.53) |
| pH | 1.06 (0.80−1.41) |
| Secchi depth (log m +1) | 0.85 (0.62−1.16) |
| Watershed agriculture ( $\sqrt{\%}$ ) | 0.81 (0.58−1.10) |
| CaO ( $\sqrt{\%}$ ) | 0.74 ** (0.57−0.92) |
| Annual temp. range (°C × 10) | 0.64 * (0.42−0.92) |
| Mean distance source (log m) | 0.61 *** (0.45−0.82) |

* $p < 0.05$, ** $p < 0.01$, *** $p < 0.001$.

### 2.4. Predicting Eurasian Watermilfoil Occurrence

We built species distribution models using logistic regression in a generalized linear modelling framework applied to the EWM occurrence dataset. The procedure employs a maximum likelihood optimization algorithm to estimate intercept and slope parameters $(\beta_0, \beta_j)$ for a set of $j$ predictors $(X)$ to determine the probability $(p)$ that a given lake $(i)$ has been invaded. The equation linearizes the response variable via a logit transformation.

$$y_i = \text{logit}(p_i) = \ln \frac{p_i}{1 - p_i} = \beta_0 + \sum_{j=1}^{n} \beta_{ij} X_{ij} \tag{1}$$

The probability is subsequently calculated as follows:

$$\text{P}(EWM \text{ presence}) = e^y / (1 + e^y) \tag{2}$$

Model fitting was performed using Firth's method of bias reduction in R using the function and package 'brglm' (v. 0.5–9) [60].

We used a 5-fold cross-validation procedure repeated 10 times to evaluate model performance. We randomly split the data into 5 approximately equal parts (termed a fold) and developed a model five times, once per each unique combination of N = 4 folds. For each run, we generated predicted values by

applying the resulting model on the remaining fifth fold [61]. After each cross-validation, we evaluated model performance. First, we conducted a receiver operating characteristic analysis as follows [62]. The logistic regression equation produces an estimate of the lake-specific probability of EWM presence. One can use the probability of presence to determine predicted presence by selecting a probability threshold above which one would assume the species is present. The threshold may be set at any point along the range in probability from 0 to 1. Starting at 0, we incrementally increased the threshold value and compared the resulting list of predicted presences to the observed presences, plotting the percentage of true presences against true absences. A 1:1 relationship between true presences and true absences would describe a model that is no better than random chance; the area under the 1:1 line is 0.50. One may plot a curve relating true absences and presences (AUC) as the threshold increases from 0 to 1. Values for the area under the curve (AUC) above 0.5 reflect predictive power that is better than chance, and a value of 1 describes a model with perfect discrimination. We generated these receiver operating characteristic curves using the function 'roc' in package 'pROC' (v. 1.9.1) [63]. We also calculated overall model deviance, the percentage of deviance explained by the model (D2), and Tjur's coefficient of discrimination, which can be interpreted much in the same way as an $R^2$ value for linear regressions [64]. We averaged all model performance statistics across the 10 repeated cross-validations.

Next, we extended the model to predict probability of presence on all lakes larger than 1 ha in surface area (N = 9285). Using the threshold values generated to evaluate the model, we selected the value 0.246 to distinguish likely presences. We used this threshold to determine which lakes were most vulnerable to EWM invasion and mapped occurrence probability using ESRI software (v. 10.2.2, Environmental Systems Research Institute, Redlands, CA, USA) [55]. Our choice of threshold allowed no more than 10% of the predicted absences to be false. Emphasizing model specificity over sensitivity minimized the chance of classifying a waterbody as "safe" when it was, in fact, vulnerable; however, see [24]. In addition, erring on the side of overpredicting may reduce bias related to survey detection failures in the occurrence data. Our approach will result in overprediction of EWM occurrence, but a less cautious prediction may be made by selecting a different threshold for the occurrence data provided in Table S2, Supplementary Materials.

*2.5. Predicting Eurasian Watermilfoil Abundance*

Next, we developed a statistical model predicting EWM abundance using the same environmental predictors on the 657 surveyed lakes. Exploratory univariate plots revealed curvilinear and unimodal distributions, so we included quadratic transformations for all predictors. Littoral EWM abundance was highly heteroskedastic, overdispersed, and right-skewed; accordingly, we selected the beta distribution for the model, which has a flexible shape controlled by mean ($\mu$) and precision ($\varphi$) parameters [65,66]. The model contains two submodels, i.e., one for the mean response and one for dispersion. We used extended beta regression models with bias correction to estimate mean and precision parameters as a function of predictors, thus modelling mean response, variable dispersion, and skewness [67]. Here, for any fixed $\mu$, greater $\varphi$ relates to decreased variability in the response variable [68]. The expected value and variance of the response variable $y$ is determined by the following:

$$E(y) = \mu \tag{3}$$

$$var(y) = \frac{\mu(1 - \mu)}{1 + \varphi} \tag{4}$$

Using a logit transformation for the mean submodel and a log link for dispersion, the models are specified as follows:

$$\text{logit}(\mu_i) = \ln \frac{\mu_i}{1 - \mu_i} = \beta_0 + \sum_{j=1}^{n} \beta_{ij} X_{ij} \tag{5}$$

$$\ln \varphi_i = \theta_0 + \sum_{j=1}^{n} \theta_{ij} Z_{ij} \tag{6}$$

The set of predictors $(X_i, Z_i)$ may vary by submodel but need not be mutually exclusive. We allowed all predictors to contribute to both submodels. The range of the response variable included 0, so we transformed it using the Smithson and Verkuilen method cited in Cribari-Neto and Zeileis [69,70]. Bias-corrected model fitting was performed using the function and package 'betareg' [69,71].

We then applied a 5-fold cross-validation procedure as described in Section 2.4. After each cross-validation, we evaluated model performance using several metrics as follows: Pearson's correlation coefficient (r) to reflect concordance between observed and predicted values, Spearman's rank correlation ($\varrho$) to test concordance among value ranks, and parameters *m* and *b* from a simple linear regression to further describe the relationship between observed and predicted values. Given perfect concordance among observed and predicted values, the intercept (*b*) and slope parameter (*m*) would be 0 and 1, indicating no bias and a comparable range of observed values at all points along the range of predicted values. Lower or higher values for *b* indicate model under- or overprediction, while a different *m* reflects a bias that may differ in magnitude along the range of predictions [72]. We calculated the root mean square error among observed and predicted values. In all cases, model performance measures were averaged across 10 repeated cross-validations. We used ESRI software to generate maps of observed and predicted abundance (v. 10.2.2, Environmental Systems Research Institute, Redlands, CA, USA) [55].

### 2.6. Defining and Prioritizing Management Targets

We extended the occurrence and abundance models developed on the 657 surveyed lakes to produce occurrence and abundance predictions for 9825 lakes over 1 ha in size. We then split lakes into categories of increasing invasion vulnerability by trisecting the ranges for predicted probability of occurrence and predicted abundance. This resulted in three groups of lakes with low, medium, and high vulnerability for EWM occurrence defined by occurrence probability thresholds at 0.496 and 0.746. For abundance, low, medium, and high vulnerability thresholds fell at 0.18 and 0.36. By cross-tabulating the priority categories for both presence and abundance, we constructed a 3-tier management priority matrix that may help direct work toward the highest-priority prevention and management targets.

## 3. Results

### 3.1. Occurrence Models

Logistic regression models predicting EWM occurrence in surveyed lakes performed well, with mean cross-validated AUC = 0.82. Variables positively related to occurrence probability included road density, surface area, maximum air temperature of the warmest month, and lake maximum depth, while factors that were negatively related included watershed % surface rock calcium oxide content, annual temperature range, and mean distance from all source populations (Table 1). Mean cross-validated deviance was 510, Tjur's coefficient of determination was 0.33, and the amount of deviance accounted for by the model was 23%.

Using a threshold probability of 0.246 to distinguish likely presences from absences resulted in a model that was 72.6% accurate in its classifications (Figure 1, Table 2). We selected the threshold probability to minimize false negatives, resulting in high model specificity. Overall, the true positive rate (sensitivity) was 0.372, while the true negative rate was 0.97 (specificity). Modeled predictions per lake are provided in Table S2 to allow alternate classifications.

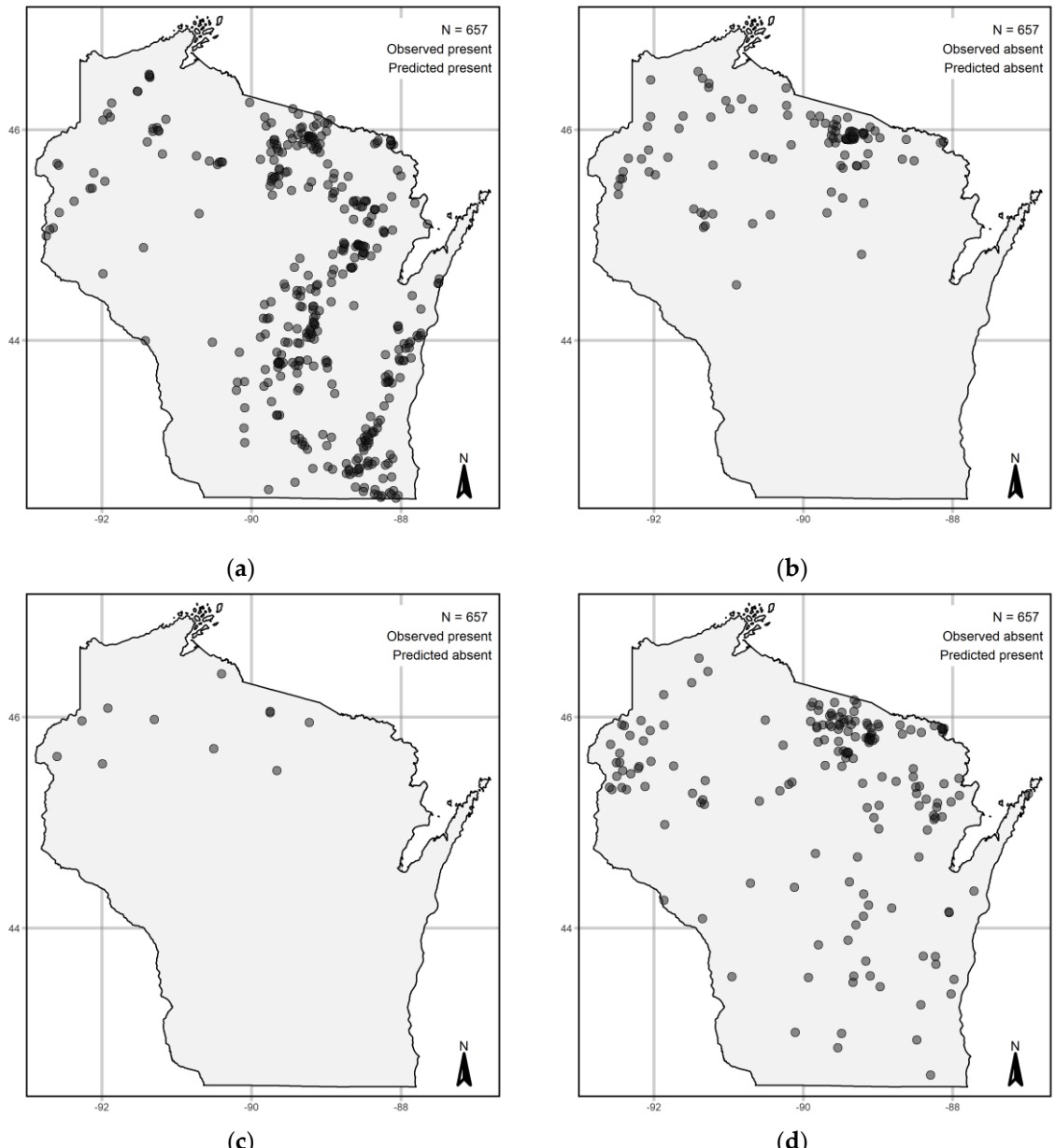

**Figure 1.** Eurasian watermilfoil (EWM) occurrence predictions for N = 657 surveyed lakes. Maps depict true presences (**a**), true absences (**b**), false absences (**c**), and false presences (**d**). Binomial response generalized linear models (GLMs) fit with Firth's bias reduction method [24].

**Table 2.** Cross-validated confusion matrix relating observed to predicted EWM occurrence in the 657 surveyed lakes. We set the threshold for classifying the predicted probability as presence at 0.246, a value that minimizes false absences to less than 10% of all predicted absences.

|  |  | Observed | |
| --- | --- | --- | --- |
|  |  | **Absent** | **Present** |
| Predicted | Absent | 100 | 11 |
|  | Present | 169 | 377 |

When the model was extended to all lakes over 1 ha in surface area, it revealed the potential for EWM to continue to expand in Wisconsin. Eurasian watermilfoil is predicted to occur in 2357 lakes where it is not currently known (Figure 1d). Vulnerable waterbodies were found statewide, but lakes

with the greatest vulnerability for occurrence tend to occur in the south-central and southeast regions of the state.

## 3.2. Abundance Models

Eurasian watermilfoil abundance models explained a small but statistically significant portion of the observed variation (log-likelihood = 1856, pseudo $R^2$ = 0.24; Table 3). Mean cross-validated deviance was −2894, root mean square error was 15.7%, while cross-validated correlation coefficients among observed and predicted abundance showed a moderate degree of concordance, with r = 0.43 and $\varrho$ = 0.41. Mean EWM abundance generally increased with conductivity, road density, and maximum air temperature, while abundance decreased with % lithological CaO and annual temperature range (Table 3). Quadratic terms indicating curvilinear relationships were important for some predictors. When all other variables were held at their mean values, abundance generally increased with increasing conductivity, then decreased at the very high end of the range. For mean distance from source, abundance first decreased, then increased, and for soil erodibility, abundance dropped at the low and high ends of the observed range, displaying a peak at intermediate values. All other predictor variables used in the model were not statistically significant. Significant precision terms were mostly negative, indicating variables associated with decreased variability in the mean response. As conductivity, alkalinity, and % watershed agriculture increased, so did the precision in the resulting estimates. Soil erodibility had a positive linear precision coefficient, while that for mean distance was negative; however, the degree to which precision changed varied across the measured range (i.e., the relationship was curvilinear).

**Table 3.** Coefficients estimated using a beta regression model for abundance data on 657 surveyed lakes. Coefficients and standard errors for mean (logit link) and precision (log-link) submodels describe patterns and variability in the EWM abundance response. Negative linear coefficients for the mean submodel indicate factors that predict lower EWM abundance, negative quadratic coefficients describe concave-down relationships, and positive quadratic coefficients are concave-up. For a fixed mean estimate in the precision submodel, a larger precision coefficient indicates lower variance in the response.

| Predictors | Mean Submodel | | | | Precision Submodel | | | |
| | Linear | | Quadratic | | Linear | | Quadratic | |
| | Estimate | SE | Estimate | SE | Estimate | SE | Estimate | SE |
|---|---|---|---|---|---|---|---|---|
| Intercept | −3.71 *** | 0.28 | | | 2.11 *** | 0.31 | | |
| Conductivity (log μS/cm) | 1.27 *** | 0.24 | −0.52 *** | 0.11 | −1.74 *** | 0.31 | 0.61 *** | 0.14 |
| Road density (log (m/ha +1)) | 0.34 * | 0.14 | −0.06 | 0.06 | −0.24 | 0.17 | 0.12 | 0.07 |
| Alkalinity (log mg $CaCO_3$ +1) | 0.31 | 0.23 | −0.01 | 0.11 | −0.73 * | 0.30 | 0.41 ** | 0.14 |
| Maximum air temp. (°C x 10) | 0.28 * | 0.11 | −0.11 | 0.10 | 0.00 | 0.12 | 0.15 | 0.11 |
| Mean distance source (log m) | 0.22 | 0.16 | 0.19 * | 0.09 | −0.30 | 0.18 | −0.23 * | 0.10 |
| Watershed agriculture ($\sqrt{\%}$) | 0.10 | 0.13 | 0.03 | 0.08 | −0.31 * | 0.15 | 0.02 | 0.09 |
| Watershed urban ($\sqrt{\%}$) | 0.10 | 0.14 | 0.02 | 0.04 | −0.23 | 0.16 | 0.00 | 0.04 |
| Maximum depth (log m +1) | 0.07 | 0.11 | −0.10 | 0.06 | 0.09 | 0.13 | 0.11 | 0.07 |
| Soil erodibility (kwfact) | 0.04 | 0.08 | −0.23 ** | 0.09 | −0.03 | 0.09 | 0.37 *** | 0.10 |
| Surface area (log ha) | 0.02 | 0.19 | 0.02 | 0.05 | 0.09 | 0.22 | −0.02 | 0.06 |
| Secchi depth (log m +1) | −0.08 | 0.10 | −0.09 | 0.06 | 0.00 | 0.11 | 0.03 | 0.06 |
| pH | −0.11 | 0.11 | 0.01 | 0.05 | 0.27 | 0.15 | −0.06 | 0.06 |
| CaO ($\sqrt{\%}$) | −0.23 * | 0.10 | 0.06 | 0.05 | 0.22 | 0.12 | −0.06 | 0.05 |
| Annual temp. range (°C x 10) | −0.54 * | 0.21 | −0.05 | 0.07 | 0.42 | 0.23 | 0.05 | 0.07 |
| Log-likelihood | 1856 | | | | | | | |
| Df | 58 | | | | | | | |
| Pseudo $R^2$ | 0.24 | | | | | | | |

* $p < 0.05$, ** $p < 0.01$, *** $p < 0.001$.

Observed and predicted EWM abundances were highly correlated (r = 0.51, ϱ = 0.52), although cross-validated performance indicated uncertainty in modelled predictions. With respect to observed abundance, predictions were relatively unbiased (*b* = −0.004) and reasonably consistent (*m* = 0.99). In the surveyed lakes, EWM abundance ranged from almost 0 to 1, with low abundances consistently observed in northern Wisconsin (Figure 2a). Predicted abundance mirrors that of observed abundance but tends to underpredict when EWM abundance is high (Figure 2b). Modelled predictions never exceeded an abundance of 0.46, but 33 out of the 657 lakes with EWM populations had observed abundance values ranging from 0.46 to 0.96.

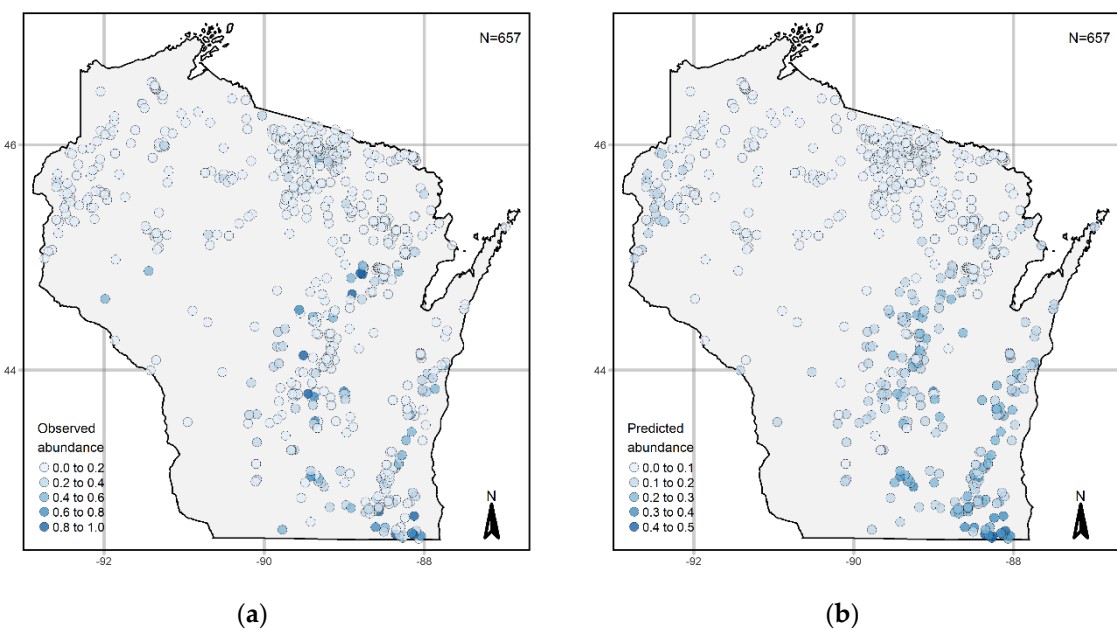

(**a**)          (**b**)

**Figure 2.** Eurasian watermilfoil abundance on N = 657 surveyed lakes. Maps depict abundance observed during aquatic plant field surveys (**a**) and predicted using beta regression (**b**).

### 3.3. Statewide Predictions

After developing the occurrence and abundance models on the N = 657 surveyed lakes, we extended model prediction to 9825 Wisconsin lakes over 1 ha in surface area (Figure 3). There was again a strong clustering of lakes at risk for abundant EWM populations in the southeast corner of the state, while northern lakes were generally less vulnerable to invasion, establishment, and abundant populations. Because occurrence is generally easier to predict on a large scale than abundance, we examined the relationship between predicted probability of occurrence and abundance, finding that predicted occurrence probability was positively related to predicted abundance (F = 0.00015, *p* < 0.001, $R_{adj}$ = 0.62), but that the relationship weakened when predicted occurrence probability was used to predict observed abundance (F = 99.07, *p* < 0.001, $R_{adj}$ = 0.13).

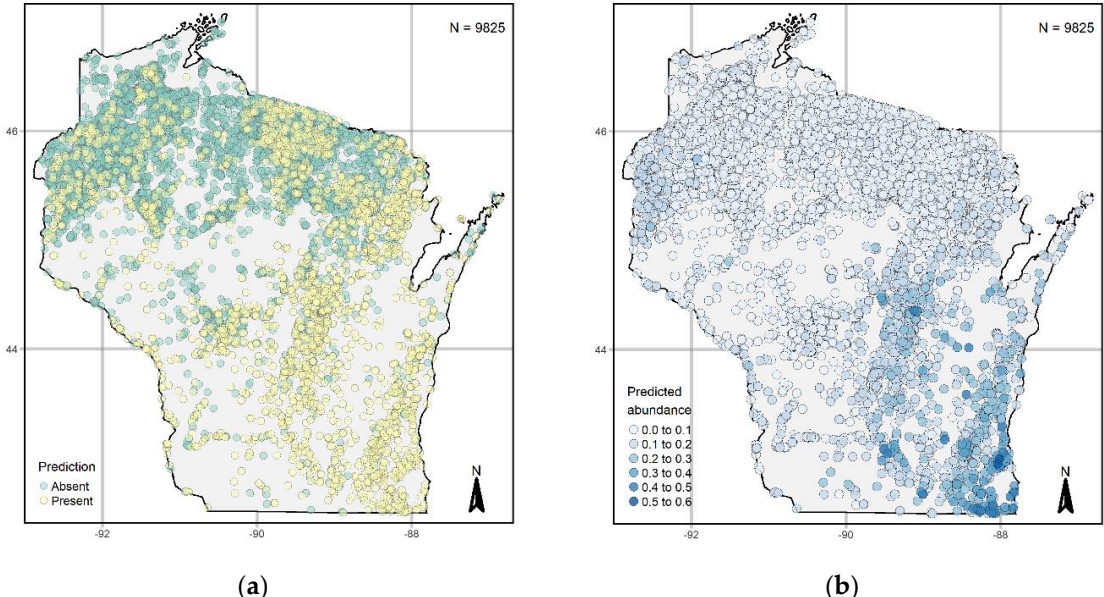

(**a**)                                                                 (**b**)

**Figure 3.** Occurrence and abundance models developed on N = 657 and extended to 9825 lakes over 1 ha in surface area. Eurasian watermilfoil occurrence predictions (**a**) generated using a bias-reduced GLM. The probability threshold distinguishing likely presences from absences was set at 0.246 to minimize the chance of false absences. Eurasian watermilfoil abundance predictions (**b**) generated using a beta regression model.

### 3.4. Prioritizing Management

We combined statewide predicted occurrence and abundance estimates to describe overall lake-specific vulnerability related to EWM's ability to arrive, survive, and thrive. We examined occurrence and abundance predictions for the 2357 uninvaded lakes where the occurrence model predicted EWM was likely to occur and assigned lakes to one of three tiers of increasing management priority (Table 4, Figure 4). Tier 3 comprises the lowest-priority lakes, including waterbodies with a relatively low occurrence risk and which are also unlikely to support abundant populations. Over 78% of lakes likely to be invaded fall into Tier 3. Of slightly higher prevention and management priority are Tier 2 lakes with a moderate risk of occurrence and abundance, a high risk of abundance balanced by a low risk of occurrence, or a high risk of occurrence balanced by a low risk of abundance. Tier 2 comprises 13% of lakes likely to be invaded. The most vulnerable lakes make up Tier 1; only 9% of lakes likely to be invaded fall into this category. In these lakes, EWM is both likely to be introduced and grow to high abundance. Tier 1 lakes occur mostly in the southeast region of the state, with vulnerability decreasing as one moves in a northwesterly direction. It is our hope that this framework along with the waterbody list presented in Table S2 can be used to inform prevention and management actions to support the efficient use of limited prevention and control resources.

**Table 4.** Three-tiered management priority matrix for uninvaded lakes that are vulnerable to EWM invasion (N = 2357 lakes where probability of occurrence is above the threshold of 0.246). Tier 1 lakes are most likely to have EWM and support abundant populations; they are assigned the top management priority. Tier 2 lakes have moderate risk for occurrence and abundance or a high risk for either occurrence or abundance. Tier 3 lakes comprise the lowest-priority group, i.e., they are still vulnerable to invasion but have a lower probability of occurrence and are less likely to support abundant populations.

| | | Abundance | | |
|---|---|---|---|---|
| | | High (0.36–1) | Med. (0.18–0.36) | Low (0–0.18) |
| Presence | High (0.75–1) | 28 | 188 | 227 |
| | Med. (0.50–0.75) | 1 | 70 | 644 |
| | Low (0.25–0.50) | 0 | 28 | 1171 |
| | Prevention and Control Priority | Tier 1 | Tier 2 | Tier 3 |

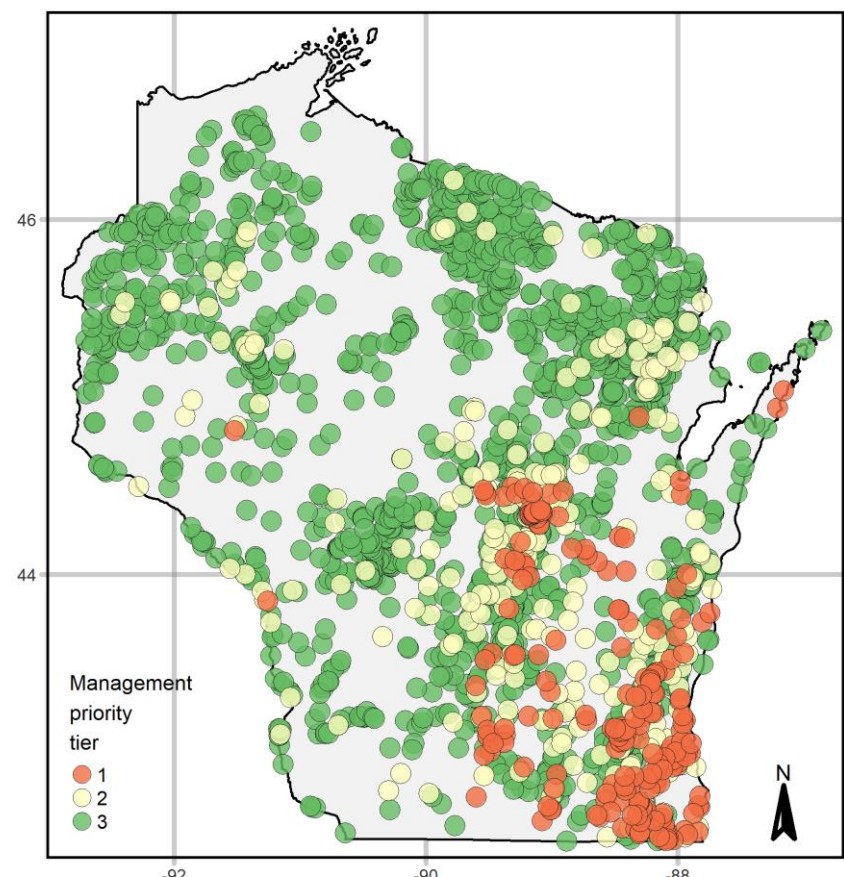

**Figure 4.** Management priority tiers based on modelled risk of EWM occurrence and abundance following the prioritization matrix in Table 4. Lakes shown are those not known to contain EWM currently, but where occurrence models predicted it is likely to occur. Tier 1 (orange) lakes are the most vulnerable. They are likely to support EWM at high abundance and have top priority for prevention and management. Tier 2 lakes include those with moderate risk for EWM invasion and abundance. Tier 3 lakes are less likely to have EWM populations and are also unlikely to support abundant populations.

## 4. Discussion

Eurasian watermilfoil has been present in Wisconsin for at least 60 years and there are still many lakes that are vulnerable to invasion [73]. However, very few vulnerable lakes are likely to support

abundant populations. While some predictors of occurrence and abundance were the same, factors related to environmental suitability, such as conductivity and soil erodibility, were uniquely predictive of abundance, whereas factors related to visitation and recreational value, such as surface area and maximum depth, emerged as unique predictors of occurrence [24,49,74]. Overall, the most vulnerable populations were found in the south and southeast regions of the state, with vulnerability decreasing northward. Lakes with a high likelihood of occurrence were not necessarily predicted to occur at high abundance, and some lakes predicted to attain high abundance did not have a high likelihood of occurrence. Our study highlights the importance of not conflating the likelihood of occurrence and abundance, which are different aspects of invasive species' success and influenced by different factors.

### 4.1. Occurrence Models

Models were developed to maximize predictive power rather than test ecological relationships, but several interesting relationships indicated by model coefficients warrant discussion. Factors associated with species transport and arrival were important predictors of EWM occurrence. Higher occurrence probability was predicted for larger and deeper lakes with more nearby roads. Larger lakes generally attract more traffic, resulting in higher propagule pressure [49]. In addition, higher spatial heterogeneity in large lakes may reduce the number of stochastic extinction events and enhance population persistence [75]. In addition, roads make lakes more accessible to humans, who are implicated as an important vector of invasive species transport [76]. Finally, invaded lakes had a smaller average distance to other established EWM populations. Dispersal probability declines with distance, such that distance from established populations is often helpful in predicting invasive species occurrence [16,77]. Distance from established populations in the occurrence model may capture constraints related to dispersal, but it may also reflect spatially auto-correlated environmental conditions that are difficult to disentangle [48].

After an invasive species arrives at a location, its survival is in part determined by local environmental conditions. Early work to model the distribution for EWM found environmental factors to be more important than those related to human activity, while a later study identified the additional importance of human activity and dispersal [23,24]. New drivers revealed by this study further add to our understanding of what controls EWM occurrence and abundance; for example, calcium oxide surface rock content predicted low occurrence probability. This variable exhibited a strong spatial pattern associated with the presence of a large dolomite deposit in eastern Wisconsin. Marl lakes occur in abundance in this region, and they have unique biogeochemical qualities. Calcium carbonate in these high-alkalinity, high-pH lakes is plentiful, but rapidly co-precipitates with phosphorus and dissolved organic material. The low concentrations of free $CO_2$, phosphorus, iron, and manganese in the water of marl lakes can limit macrophyte growth [78]. Maximum air temperature was positively associated with EWM occurrence, whereas annual temperature range was negatively related. Eurasian watermilfoil can grow in a wide range of temperatures with populations extending up to 68.8°N latitude [79]. Temperature is unlikely to be a limiting factor for this species in the spatial extent considered by this study. In addition, these climate factors may be co-linear with a set of uncaptured variables related to environmental conditions or invasion history.

Other environmental factors such as conductivity, alkalinity, and water clarity were not significant predictors of EWM occurrence. This may be due to the species' broad environmental tolerances and the fact that environmental variables were mostly within published tolerance ranges [28]. That said, dissolved inorganic carbon is an important nutrient source for the species; alkaline lakes are generally considered more suitable. In our dataset, EWM never occurred when conductivity was below 16 μS/cm or when alkalinity was below 4 mg $CaCO_3$/L. Around 6% of lakes statewide had conductivity below 16μS/cm and 7.5% had alkalinity below 4 mg $CaCO_3$/L. Additional work is necessary to determine whether this threshold is biologically relevant for EWM or an artifact of the distribution of values within our sample.

Species occurrence models typically assume a population is at equilibrium, but this assumption is violated in the case of a range-expanding species like EWM. The inclusion of true absence data along with spatial variables related to dispersal allows us to reduce the bias that would otherwise be present [16]. Still, it is likely that our training set includes occurrences that have not been detected, leading to some degree of error in our predictions. While the extrapolated statewide predictions we present can be useful in planning prevention and management activities, it is important to treat them as indicators of a likely but uncertain future state.

*4.2. Abundance Models*

Once probability of introduction and survival is known, the remaining question is one of impact: if the species is introduced and survives, is it also likely to cause problems? While impact is arguably the most important filter to consider, it is often the most difficult to predict [20]. Eurasian watermilfoil has been associated with several effects on native flora, macroinvertebrates, habitat, and water quality [35,80–83]. Eurasian watermilfoil has also been associated with recreational impairment and decreased lake property value [37,84,85]. That said, the magnitude of its socioeconomic and ecological effects are most likely directly related to abundance [19].

In contrast to the EWM occurrence model, several local environmental variables were uniquely predictive of abundance (Table 3). Conductivity was positively associated with *M. spicatum* abundance. Centered and standardized predictors allow model coefficients to be interpreted as effect sizes, and conductivity had one of the largest effects observed. The influence of conductivity is strong; it is often considered a "master factor" driving landscape-scale species distributions [74]. In particular, the strong effect makes sense for EWM, which can use bicarbonate as a source of carbon dioxide, thus attaining a competitive advantage in high-conductivity and high-alkalinity lakes [86].

Soil erodibility displayed a concave-down curvilinear relationship such that lakes in watersheds with moderate soil erodibility supported more abundant populations. EWM is tolerant of nutrient enrichment and prefers moderately enriched lakes [28]. Phosphorus in surface water is often derived from rock and soils and is generally higher when watersheds are comprised of highly erodible soil [87]. Erodible soil is also often favored for agriculture, where exogenous additions of fertilizer further enrich surface waters [88]. As nutrient content in water increases, so does primary productivity; abundant plant populations are often found in enriched waters. However, beyond a certain level of enrichment, filamentous and planktonic algae begin to dominate, resulting in water with low light transmission. Macrophytes then decrease in abundance, unable to survive in these turbid, highly enriched waters [89,90].

Temperature, calcium oxide surface rock content, and distance from established populations were significant predictors of both occurrence and abundance. Higher air temperature and lower annual temperature range were associated with higher abundance. Eurasian watermilfoil has a relatively high optimum temperature for photosynthesis, and warmer temperatures likely enhance productivity and expansion [91]. Surprisingly, as distance from invaded systems increased, so did abundance. This may be explained by a relationship between distance from established populations and time since introduction—reflecting the "boom" typically exhibited by EWM populations preceding a decline that often occurs around 10–15 years following invasion [31,92,93].

*4.3. Management Prioritization: Uniting Occurrence and Abundance*

Many lakes are vulnerable to EWM introduction, but of the 2357 lakes in which EWM is likely to occur, only 245 have the highest priority for prevention and management given their combined risk of occurrence and abundance. Moreover, of the 245 Tier 1 lakes singled out for action, only 29 (12%) are in the highest abundance risk category. This makes sense, given that we know species abundance distributions for non-native and native species are similar—they are "commonly rare and rarely common" [94]. This also aligns with our understanding of EWM abundance distributions—of 388 surveyed lakes with EWM populations, only 50 (13%) are in the highest abundance category

(>0.36). In light of our predictions, we recommend enhancing monitoring and prevention programs on the 29 highest-priority lakes to help offset large future costs and improve management efficiency [34]. The remaining Tier 1 lakes that are vulnerable to EWM introduction but which have a lower risk for abundant populations may present opportunities for public engagement through community-based science and prevention.

A key implication of our study is that even though EWM invaded Wisconsin decades ago, there remain many uninvaded lakes that are predicted to be suitable for establishment, but significantly fewer lakes where the species is expected to reach high abundance. Given this finding, these models may be an important tool for prioritizing resources. Considering that the species has been present and spreading for decades, one might expect EWM to be largely saturated on the landscape, having already established itself in most suitable lakes. The lesson here applies to other invasive species—just because an invader has been long present in a region does not mean that it has established itself in all suitable habitat. There is value in efforts to prevent or slow further spread, even for long-established non-native species.

Management of EWM can be costly. Placing equal priority on all lakes fails to take advantage of the fact that, for most lakes, EWM abundance is low. Yet managers are justifiably risk-averse, and proactive management to keep populations small in case the population can attain high abundance is understandable. The approach we present here, to combine occurrence and abundance predictions, allows researchers and lake managers to empirically determine a management strategy appropriate for a recent introduction based on evidence; a wait-and-see approach may be appropriate when the likelihood of achieving abundant populations is low, while a more aggressive or proactive approach would make sense when the predicted abundance is high. Better justification for planning a proactive versus reactive strategy will hopefully result in more efficient allocation of limited management funding.

Understanding multiple levels of the invasion process helps generate realistic predictions of system-specific risk [11,95]. Few efforts to date have produced risk assessments that integrate occurrence and abundance, but see [19]. We could improve on this assessment of vulnerability with a more nuanced understanding of species' impacts across systems. We selected risk thresholds to determine management priority by simply trisecting the observed range in predicted occurrence and abundance, assuming adverse effects have a simple positive linear relationship to abundance. An improvement to this work would be to select prioritization thresholds in consideration of the actual shape of the impact–abundance relationship(s) [19,21,22].

A final caveat in using these predictions to direct management effort is the following: the EWM occurrence model explained 32% of observed variation and correctly classified 75.6% of surveyed lakes. The abundance model explained 25% of the observed variation. At a statewide scale, occurrence and abundance predictions allow strategic use of limited resources despite the uncertainty in the model. However, when evaluated at the individual lake scale, error in the model should be recognized to appropriately characterize risk. The most conservative option for a less vulnerable Tier 3 lake with energetic and willing volunteers may still be to support active prevention.

In conclusion, understanding lake-specific vulnerability in light of predicted occurrence and abundance is a promising approach to effective management of AIS across a landscape. Prevention and control funding and volunteer effort are precious resources that should be directed to the most vulnerable waterbodies. Smart prevention and prudent control guided by a clear prioritization framework will save money, minimize non-target effects, and hopefully result in better management outcomes.

**Supplementary Materials:** The following are available online at http://www.mdpi.com/1424-2818/12/10/394/s1, Table S1 summarizes the variables used to predict EWM occurrence and abundance. Table S2 displays observations and modelled predictions for EWM occurrence and abundance for all 9287 study lakes. Also presented are the final management priority tier assigned by the prioritization framework and the relative lake-specific risk for EWM occurrence and high abundance. Lakes are identified by the WDNR waterbody identification code, lake name, and county.

**Author Contributions:** A.M. and E.R.K. conceptualized the work, A.M. and M.J.V.Z. designed the study, A.M. assembled and analyzed the data and designed visualizations, A.M. wrote the original draft. A.M., C.L.H., S.V.E.,

E.R.K. and M.J.V.Z. reviewed and edited subsequent drafts. A.M. and M.J.V.Z. secured funding for the work. All authors have read and agreed to the published version of the manuscript.

**Funding:** This research was funded by the National Science Foundation through the Graduate Research Fellowship Program under Grant #DGE-1256259 and the North Temperate Lakes Long-Term Ecological Research Program under cooperative agreement #DEB-1440297, and the Wisconsin Department of Natural Resources.

**Acknowledgments:** We would like to thank Paul Frater for his review of our mathematical notation and the staff at the Wisconsin DNR for their help collecting data, including Therese Ashkenase, Martha Barton, Dana Bigham, Shaunna Chase, Michael Fell, Paul Frater, Elizabeth Haber, Raffica La Rosa, Michelle Nault, Meghan Porzky, Erin Ridley, Chris Repking, Katie Roth, Jesse Schwingle, Nicholas Shefte, Kari Soltau, and Kelly Wagner. We thank DNR central and regional staff for supporting and participating in this work, including Tim Asplund, Heidi Bunk, Mary Gansberg, Kevin Gauthier, Susan Graham, Ted Johnson, Jim Kreitlow, Brenda Nordin, Scott Provost, Carroll Schaal, Alex Smith, and Pamela Toshner.

**Conflicts of Interest:** The authors declare no conflict of interest.

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
