# Peer review of "Prioritizing Management of Non-Native Eurasian Watermilfoil Using Species Occurrence and Abundance Predictions"

_diversity, doi:10.3390/d12100394_

Round 1

Reviewer 1 Report

The manuscript “Prioritizing management of non-native Eurasian watermilfoil using species occurrence and abundance predictions” constitutes an interesting study to provide a way to identify lakes that need to be monitored more closely for the presence of EWM. The manuscript is well described, and the statistical analysis is appropriate. However, as the authors themselves asserted, the significance of the models was rather low. This means that other predictors should be considered in future studies to describe the variability of the phenomenon with greater accuracy. Despite this, the approach is correct and, therefore, the manuscript can be accepted after minor revision.

Minor comments/suggestions follow.

Page 1: Keywords: please, eliminate acronyms EWM and SDM

Page 3: At the end of introduction the authors wrote “function of water quality, land use, dispersal potential, geology and climate. Predictors included local (e.g. water conductivity, water clarity, nearshore urban development) and regional factors (e.g. annual temperature range, watershed land use)“. The authors should explain why these predictors were chosen and if there are previous studies that correlate these predictors to EWM.

Page 6: The authors should explain why the explanatory variables were chosen and if there are previous studies that correlate these variables to EWM.

Page 10: How were the values for occurrence probability thresholds of <0.496 and 0 .746 chosen?

Page 11: Please, define AUC

Table 1 and table 3: are the authors sure about (√%) for Watershed urban, Watershed agriculture and CaO?

Page 15: the authors wrote “Higher occurrence probability was predicted for larger, deeper lakes with more nearby roads”. However, no data was reported for this statement. In fact, since the beginning of the study I wondered if there was a correlation between the presence of macrophytes and the size of the lakes. In the model results in table 3, it appears the maximum depth and surface area predictors are not significant. This is in contradiction with what the authors said.

Author Response

Thank you for your review! Your comments have improved the quality of the manuscript.

Page 1: Keywords: please, eliminate acronyms EWM and SDM

We have used “Eurasian watermilfoil” and “Species Distribution Model” throughout. We have kept “EWM” and “SDM” as keywords due to their widespread usage.

Page 3: At the end of introduction the authors wrote “function of water quality, land use, dispersal potential, geology and climate. Predictors included local (e.g. water conductivity, water clarity, nearshore urban development) and regional factors (e.g. annual temperature range, watershed land use)“. The authors should explain why these predictors were chosen and if there are previous studies that correlate these predictors to EWM.

We added a general explanation to page 3 and include citations that cover the relationship between predictors, EWM suitability, and invasive species occurrence and spread.

Predictors were selected for their link to environmental suitability for Eurasian watermilfoil and aquatic invasive species occurrence (Smith and Barko 1990, Buchan and Padilla 2000, Mikulyuk et al. 2011, Decker et al. 2017).

Page 6: The authors should explain why the explanatory variables were chosen and if there are previous studies that correlate these variables to EWM.

We add supporting citations for each variable in Supporting Information Table S1.

Page 10: How were the values for occurrence probability thresholds of <0.496 and 0 .746 chosen?

We clarified in the text that these thresholds follow trisecting the range of predicted probabilities:

We then split lakes into categories of increasing invasion vulnerability by trisecting the ranges for predicted probability of occurrence and predicted abundance. This resulted in three groups of lakes with low, medium, and high vulnerability for Eurasian watermilfoil occurrence defined by occurrence probability thresholds at 0.496 and 0.746…

Page 11: Please, define AUC

“Area under the curve”, text added to page 8.

Table 1 and table 3: are the authors sure about (√%) for Watershed urban, Watershed agriculture and CaO?

Yes, we square-root transformed skewed percentages. This is briefly described in the methods on page 7.

Page 15: the authors wrote “Higher occurrence probability was predicted for larger, deeper lakes with more nearby roads”. However, no data was reported for this statement. In fact, since the beginning of the study I wondered if there was a correlation between the presence of macrophytes and the size of the lakes. In the model results in table 3, it appears the maximum depth and surface area predictors are not significant. This is in contradiction with what the authors said.

We believe there is some confusion in aligning the text and tables for the occurrence and abundance models. Table 3 presents coefficients for the abundance model, in which depth and size were not significant.  Table 1 displays the coefficients for the occurrence model, in which maximum depth and surface area were statistically significant.

Reviewer 2 Report

Review for Mikulyuk et al.

Generally I really like the approach the authors took. It’s a great way to use presence/absence data to build likely occurrence and abundance models for species. It is especially useful for aquatic species that have discrete borders in closed bodies of water.

Since the manuscript did not have any page or line numbers, I made my suggested edits and asked my questions within the submitted pdf.

All in all, I think the authors did a nice job of presenting the problem, applying the research and models and coming up with an easily actionable plan for lake managers.

Author Response

Thank you for your review! Your comments have improved our manuscript and we are excited to move forward to publication.

Page 1

Suggest: previously unimpacted

We deleted “additional” to leave “many lakes are still vulnerable,” instead. We think the fact that the lakes are previously unimpacted is now more clearly implied.

Page 2

Space deleted, “to” deleted, “of” added

Page 3

“Solution” accepted

Page 4

Deleted redundancy, accepted “$500,000 USD”

Page 4

We took this suggestion with a slight modification: “Eurasian watermilfoil management can also have significant non-target effects.” 

“Such as” accepted. Deletion of “the species” rejected, but “and” added for clarity.

We replaced “is critical” with “is a priority” and accepted modifications to the last sentence.

Page 5

Citation for determining the number of points added.

Page 6

There's lots of emphasis on roads and size of lakes and other human-favored factors, but you never explicitly state that lake access matters. Could public ramps be added or some other factor of access to improve accuracy? Or did I miss that in the parameters somewhere?

Regional biologists report the DNRs public access database is marginally accurate, so we chose not to include boat/canoe launches. Instead, we use road density and lake size as a proxy for access and recreational boat traffic. This link is demonstrated by Reed-Anderson et. al 2000 and Decker et al.  2017. We added these citations to the methods text.

Page 12

We edited the hanging participle:

Soil erodibility had a positive linear precision coefficient, while that for mean distance was negative, however the degree to which precision changed varied across the measured range (i.e. the relationship was curvilinear).

Page 16

Deletion accepted

Page 18

Edits to first sentence in section 4.3 accepted. 2 additional re-wording suggestions accepted with slight changes:

Enhancing monitoring and prevention programs on the 29 highest-priority lakes could help offset large future costs and improve management efficiency (Zipp et al. 2019). The remaining tier 1 lakes that are vulnerable to Eurasian watermilfoil introduction but which have a lower risk for high abundance may present opportunities for public engagement through community-based science. (aside: we prefer to separate the idea of citizenship from volunteer science)

“Critically important” replaced with “important”, other changes accepted.

Page 19

“present and” added, “expect” accepted

We agree the prioritization can be used to enhance screening and monitoring – we tried to make that point clearer in the prior paragraph.

Page 20

“at” accepted

Page 21

Edited citations.

Page 32

Yes, the model over-predicts false positives—a function of our ‘conservative’ decision to be nearly intolerant of false absences. Others could decide to balance the specificity and sensitivity differently. We hope to support this opportunity by providing the data in the supplementary material. We made this point clear at the end of section 2.4

“(d)” inserted.